# A phase 1 study of nivolumab in combination with interferon-gamma for patients with advanced solid tumors

Matthew Zibelman [1] ✉, Alexander W. MacFarlane IV[2], Kimberly Costello[3], Thomas McGowan[3], John O'Neill[3], Rutika Kokate [3], Hossein Borghaei [1], Crystal S. Denlinger[1], Efrat Dotan[1], Daniel M. Geynisman[1], Angela Jain[1], Lainie Martin[1], Elias Obeid[1], Karthik Devarajan[4], Karen Ruth[4], R. Katherine Alpaugh[5], Essel Al-Saleem Dulaimi[6], Edna Cukierman [7], Margret Einarson[8], Kerry S. Campbell[2] & Elizabeth R. Plimack [1]

This phase I, dose-escalation trial evaluates the safety of combining interferon-gamma (IFN-γ) and nivolumab in patients with metastatic solid tumors. Twenty-six patients are treated in four cohorts assessing increasing doses of IFN-γ with nivolumab to evaluate the primary endpoint of safety and determine the recommended phase two dose (RP2D). Most common adverse events are low grade and associated with IFN-γ. Three dose limiting toxicities are reported at the highest dose cohorts. We report only one patient with any immune related adverse event (irAE). No irAEs ≥ grade 3 are observed and no patients require corticosteroids. The maximum tolerated dose of IFN-γ is 75 mcg/m$^2$, however based on a composite of safety, clinical, and correlative factors the RP2D is 50 mcg/m$^2$. Exploratory analyses of efficacy in the phase I cohorts demonstrate one patient with a complete response, and five have achieved stable disease. Pre-planned correlative assessments of circulating immune cells demonstrate intermediate monocytes with increased PD-L1 expression correlating with IFN-γ dose and treatment duration. Interestingly, post-hoc analysis shows that IFN-γ induction increases circulating chemokines and is associated with an observed paucity of irAEs, warranting further evaluation. ClinicalTrials.gov Trial Registration: NCT02614456.

Systemic immune checkpoint blockade (ICB) with drugs targeting the programmed death one (PD-1) pathway has revolutionized oncology. ICB offers the potential for some patients to experience durable anti-tumor responses; however, the majority of patients do not respond to single-agent ICB[1]. Numerous combinatorial approaches are exploring whether combining additional agents with PD-1 pathway inhibitors can increase the quantity or quality of responses. For example, combined ICB with drugs targeting both the PD-1 pathway along with the cyto-toxic T-lymphocyte-associated protein 4 (CTLA-4) checkpoint has demonstrated increased efficacy in several cancers, albeit with

[1]Department of Hematology Oncology, Fox Chase Cancer Center, Philadelphia, PA, USA. [2]Immune Monitoring/Cell Sorting Facility, Institute for Cancer Research, Philadelphia, PA, USA. [3]Office of Clinical Research, Fox Chase Cancer Center, Philadelphia, PA, USA. [4]Biostatistics and Bioinformatics Facility, Fox Chase Cancer Center, Philadelphia, PA, USA. [5]Protocol Support Laboratory, Fox Chase Cancer Center, Philadelphia, PA, USA. [6]Department of Pathology, Fox Chase Cancer Center, Philadelphia, PA, USA. [7]Cancer Signaling and Microenvironment Program, Marvin and Concetta Greenberg Pancreatic Cancer Institute, Fox Chase Cancer Center, Philadelphia, PA, USA. [8]High Throughput Screening Facility, Fox Chase Cancer Center, Philadelphia, PA, USA. ✉e-mail: matthew.zibelman@fccc.edu

increased toxicity[2–4]. Multiple mechanisms have been proposed for the improved efficacy of combined ICB, including the release of non-overlapping co-inhibitory pathways and the activation of complementary anti-tumor T-cell subsets[5]. Discovering novel strategies to optimize ICB in tumors less likely to respond or that have acquired resistance to single agent PD-1 inhibition, without increasing immune-related adverse events (irAEs), are needed.

Cytokines have been used for decades as anti-cancer therapeutics with mixed success. Interferon-gamma (IFN-γ) is the only type II interferon and is FDA-approved for the treatment of two rare pediatric diseases, osteopetrosis and chronic granulomatous disease. In the 1980s-90s it was evaluated as a single agent for use against a variety of different cancers and demonstrated limited activity[6–8]. One potential explanation for its lack of success as a single agent may be the confounding impacts of immune checkpoints, particularly the PD-1 ligand (PD-L1), which had not been discovered when these trials were conceived. Since IFN-γ directly up-regulates PD-L1, the presence of IFN-γ in the tumor microenvironment (TME) may enhance an immunosuppressive milieu, further promoting T-cell inhibition and creating an immune inhibitory environment favoring tumor progression[9]. However, IFN-γ also has a multitude of pro-inflammatory properties, including acting as a chemoattractant, up-regulating expression of major histocompatibility complexes (MHC), controlling immunoproteasome machinery to enhance antigen presentation, augmenting interactions between macrophages and T-cells, and regulating differentiation of T-cell effector subsets[10,11]. Thus, we hypothesized that the pro-inflammatory properties of IFN-γ therapy could prime an effective anti-tumor response, and that adding a PD-1 pathway-targeted monoclonal antibody (mAb) would ablate the associated inhibitory impacts of PD-L1 upregulation to facilitate more robust and sustained T-cell mediated tumor cell destruction.

Here we report the results of a phase I dose escalation trial of the combination of IFN-γ and the anti-PD-1 targeted mAb nivolumab in patients with advanced solid tumors (NCT02614456). The trial establishes a recommended phase two dose of IFN-γ at 50 mcg/m² and is well tolerated, with one patient experiencing a durable complete response and five patients achieving stable disease as best response. No patients developed a serious immune-related adverse event (irAE), and post-hoc exploratory correlative chemokine analysis supports a hypothesis warranting further study as to whether IFN-γ may promote an environment restraining irAE development.

## Results
### Patient demographics
From December 29, 2015 to February 12, 2018, 26 patients were accrued to four different dose cohorts of IFN-γ. Patient demographics are shown in Table 1. The median age was 60 years and 16 patients (61.5%) were female. The most common tumor types included renal cell carcinoma (RCC), gastroesophageal carcinoma (GEC), triple-negative breast cancer (TNBC), and ovarian carcinoma. Twelve patients (46.2%) had previously progressed while receiving a checkpoint inhibitor targeting the PD-1 pathway.

### Evaluable patients
Of the 26 patients accrued, all were evaluable for the primary safety analysis, and 23 were evaluable for exploratory efficacy analysis. Three patients in the 25 mcg/m² IFN-γ dose cohort were unable to complete the DLT phase and two were replaced. One patient had worsening myalgias as an exacerbation of his chronic hypertrophic pulmonary osteoarthropathy while on IFN-γ induction and withdrew consent before the first dose of nivolumab. Another patient developed a new pleural effusion during IFN-γ, which was attributed to the IFN-γ but occurred prior to starting the nivolumab, thus, that patient was not able to complete the DLT period and was replaced. The final patient accrued to the 25 mcg/m² fourth cohort came off for disease progression prior to completing the

### Table 1 | Patient characteristics

| Characteristic | All Pts | Prior IO |
|---|---|---|
| Pts evaluable for safety, n | 26 | 12 |
| Pts Evaluable for Efficacy, n | 23 | 9 |
| Median age, yrs (min–max) | 60 (33-76) | – |
| *Gender, n (%)* | | |
| Female | 16 (61.5%) | – |
| Male | 10 (38.5%) | – |
| *Race, n (%)* | | |
| Caucasian | 22 (84.6%) | – |
| Black, Asian, other | 4 (15.4%) | – |
| *ECOG performance status, n (%)* | | |
| PS 0 | 7 (26.9%) | – |
| PS 1 | 19 (73.1%) | – |
| *Tumor type, n (%)* | | |
| Renal cell carcinoma (RCC) | 5 (19.2%) | 4 |
| Gastroesophageal (GE) | 4 (15.4%) | 2 |
| Triple neg breast (TNBC) | 4 (15.4%) | 0 |
| Ovarian | 4 (15.4%) | 1 |
| Endometrial | 3 (11.5%) | 0 |
| Lung | 3 (11.5%) | 3 |
| Other (anal, mesothelioma, urothelial) | 3 (11.5%) | 2 |
| Median number of prior therapies (Min–Max) | 4 (1 -15) | – |
| Received prior immunotherapy, n (%) | 12 (46.2%) | |

Baseline characteristics of all patients on trial evaluable for safety and efficacy. The second column includes only those patients that received prior immune checkpoint blockade before going on this trial.
*IO* immunotherapy, *Pts* patients.

DLT phase; however, this patient was not replaced as no DLTs were recorded in other patients in this cohort, thus that dose would be considered safe regardless. One patient in the 50 mcg/m² cohort of IFN-γ went off trial during the DLT phase for the progression of the disease and was replaced. One patient in the 100 mcg/m² cohort had a DLT prior to the first response evaluation and was unevaluable for efficacy.

### Safety
Adverse events are summarized in Table 2 and supplementary Tables 1, 2 There were no grade 5 AEs. The most common AEs irrespective of grade were known IFN-related side effects such as fatigue, fever, chills, myalgias, and headache. These were predominantly <grade 3, attributable to IFN-γ, and were generally manageable with acetaminophen and supportive care. Elevations in either hepatic transaminases attributable as at least possibly related to either drug occurred in 57.7% of patients (aspartate aminotransferase [AST] only in 15.3%, alanine aminotransferase [ALT] only in 3.8%, and both in 38.4%). Most were grade 1 and transient, except two patients had transient grade 2 elevations in AST and one patient had a grade 3 elevation in AST that was denoted a DLT. Lymphocyte count reductions occurred in > 30%, with 19.2% reaching grade 3-4 and attributed at least possibly to either drug. No patients developed neutropenic fever and neutrophil levels improved once IFN-γ was stopped. Two patients discontinued the trial due to a drug-related adverse event—one due to pleural effusion and one for myalgias, both attributed to IFN-γ. Three patients skipped one dose of nivolumab for AEs the day of treatment, two of which were ultimately determined to be disease-related clinical declines, and the third was due to elevations in liver enzymes discussed below.

There were three DLTs on the trial, one in the 75 mcg/m² cohort, and two in the 100 mcg/m² cohort. The patient in the 75 mcg/m² cohort developed grade 3 fatigue, possibly related to both drugs, that

**Table 2 | All-grade adverse events occurring in >2 patients**

| Adverse Event Term | Any Grade 1 to 4 | All, n = 26 | | IFN 25 mcg/m², n = 5 | | IFN 50 mcg/m², n = 8 | | IFN 75 mcg/m², n = 7 | | IFN 100 mcg/m², n = 6 | |
|---|---|---|---|---|---|---|---|---|---|---|---|
| | n (%) | Grade 1-2 | 3+ | Grade 1-2 | 3+ | Grade 1-2 | 3+ | Grade 1-2 | 3+ | Grade 1-2 | 3+ |
| Fatigue | 20 (76.9) | 16 | 4 | 5 | | 3 | 2 | 4 | 1* | 4 | 1* |
| Fever | 15 (57.7) | 15 | | 3 | | 3 | | 5 | | 4 | |
| Aspartate aminotransferase increased | 14 (53.8) | 13 | 1 | 2 | | 4 | | 6 | | 1 | 1* |
| Myalgia | 14 (53.8) | 14 | | 5 | | 2 | | 5 | | 2 | |
| Anorexia | 13 (50.0) | 13 | | 2 | | 3 | | 3 | | 5 | |
| Chills | 13 (50.0) | 13 | | 2 | | 3 | | 5 | | 3 | |
| Headache | 13 (50.0) | 13 | | 2 | | 4 | | 5 | | 2 | |
| Alanine aminotransferase increased | 11 (42.3) | 11 | | 1 | | 3 | | 4 | | 3 | |
| White blood cell decreased | 9 (34.6) | 8 | 1 | 1 | | 1 | | 5 | 1 | 1 | |
| Anemia | 8 (30.8) | 6 | 2 | 2 | | 1 | 1 | 2 | | 1 | 1 |
| Lymphocyte count decreased | 8 (30.8) | 7 | 1 | 1 | | 3 | | 2 | 1 | 1 | |
| Arthralgia | 7 (26.9) | 7 | | 2 | | 1 | | 2 | | 2 | |
| Weight loss | 7 (26.9) | 7 | | | | 3 | | 1 | | 3 | |
| Intermittent sweats | 6 (23.1) | 6 | | 1 | | 2 | | 3 | | | |
| Alkaline phosphatase increased | 5 (19.2) | 5 | | 1 | | 2 | | | | 2 | |
| Nausea | 5 (19.2) | 5 | | | | 2 | | 2 | | 1 | |
| Diarrhea | 4 (15.4) | 4 | | | | 2 | | 1 | | 1 | |
| Dyspnea | 4 (15.4) | 4 | | 2 | | | | 1 | | 1 | |
| Hot flashes | 4 (15.4) | 4 | | 2 | | | | 1 | | 1 | |
| Hyponatremia | 4 (15.4) | 2 | 2 | | | | 1 | 1 | | 1 | 1 |
| Neutrophil count decreased | 4 (15.4) | 4 | | | | | | 4 | | | |
| Dizziness | 3 (11.5) | 3 | | | | 1 | | 2 | | | |
| Pain in extremity | 3 (11.5) | 3 | | 1 | | 1 | | 1 | | | |
| Pleural effusion | 3 (11.5) | 1 | 2 | 1 | | | | | | 1 | 1* |

All grade adverse events occurring in > 2 patients attributed as at least possibly related to either drug and stratified by dose level and grade (* denotes dose limiting toxicity).

met criteria for DLT and was taken off trial. In the 100 mcg/m² cohort, one patient developed a grade 3 pleural effusion and grade 3 dyspnea possibly related to both drugs, while a second patient experienced grade 3 elevation in liver enzymes that met criteria for DLT. The patient with dyspnea and effusion went off trial and died within 4 months. While grade 3 elevation in liver function tests was attributed as possibly related to nivolumab, that patient had hepatic metastases and evidence for clinical progression of disease and ultimately died of disease progression shortly after discontinuing on study. The MTD of IFN-γ was determined to be 75 mcg/m².

Two AEs of interest occurred during the trial. In the 50 mcg/m² cohort, two patients developed symptomatic brain metastases while on trial (mRCC and TNBC). Neither patient had brain imaging prior to starting on trial, thus subsequent cohorts required brain imaging for trial inclusion. One patient with metastatic RCC and stable systemic disease in the 25 mcg/m² cohort died of a hemorrhagic stroke from a metastatic lesion despite a negative baseline MRI. Seven patients (26.9%) on trial developed new or worsening ascites or pleural effusion while on treatment, including two patients that came off trial for new pleural effusions (one attributed to IFN-γ, one possible but ultimately due to progression). One patient noted above developed a new, large pleural effusion during the one-week induction and prior to receiving the first nivolumab dose. Otherwise, all remaining patients developing fluid accumulation had either a malignancy susceptible to ascites (e.g., ovarian cancer) or known baseline disease in the pleura or peritoneum that worsened on treatment (Supplementary Table 3).

We also looked at irAEs that would be normally associated with treatment with nivolumab; however, there were no grade ≥ 3 irAEs and no patients received steroids for management of any irAE. The only AE of any grade attributable as an irAE occurred in one patient. This was a patient with TNBC who experienced a complete response (CR) on trial and completed the 3 months of IFN-γ and the entire year of nivolumab. While on nivolumab alone, the patient experienced grade 2 arthralgias that started three months after completing IFN-γ. This was managed with anti-inflammatory supportive medications initially, later maintained on a tricyclic antidepressant.

**Efficacy**
Twenty-three patients were evaluable for post-hoc exploratory efficacy analysis (Fig. 1a, b). Overall, one patient experienced a CR (4.3%), five patients achieved stable disease (SD) as best response (21.7%), and the remaining patients had primary progressive disease. The ORR was 4.3% and the disease control rate (DCR; DCR = CR + PR + SD) was 26.1%. The median PFS was 3.0 months (95%CI 2.0–3.3) [Supplementary Fig. 1] and the median OS was 7.9 months (95%CI 5.6–15.4) [Supplementary Fig. 2]. Overall survival at 12 months was 39.1% (95% CI = 19.9–58.0%). Among evaluable patients with a diagnosis of RCC (n = 4), three of four patients had stable disease, including two who had received prior ICB. Among evaluable patients who had received prior immunotherapy (n = 9, 39.1%), 2 achieved stable disease. Of the five patients achieving stable disease, three had a duration of treatment lasting less than six months. One patient with an esophageal carcinoma and prior immunotherapy achieved progressive disease as best response, however remained on treatment for 40 weeks with clinical benefit.

**Tumor biopsy analysis**
Planned secondary analyses of baseline tumor biopsies were compared to on-treatment biopsies collected after IFN-γ induction but

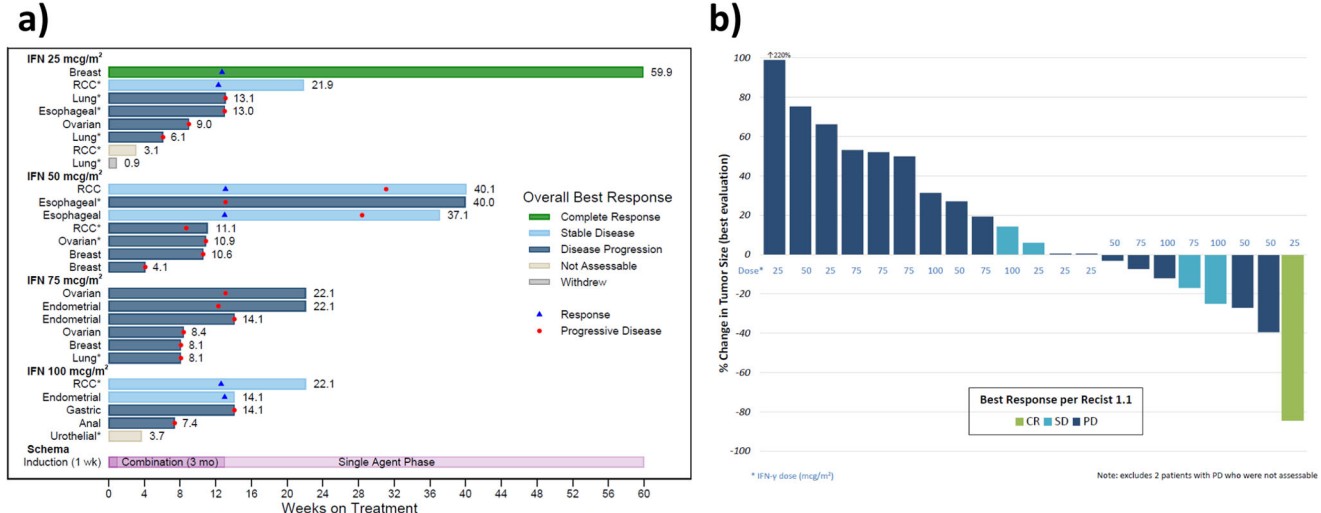

**Fig. 1 | Efficacy. a** Swimmer's Plot. Denotes duration of treatment of each individual patient in weeks, starting from the first dose of treatment at induction until coming off trial-specified therapy, separated by dose cohort and labeled by tumor type. Best response is denoted by color bar. Asterisks denote patients that had received a prior immune checkpoint inhibitor. A red dot denotes time of documented disease progression, while the blue triangles represent time of best response. Source data are provided as a Source Data file. mcg= micrograms; RCC= renal cell carcinoma.

*denotes prior immunotherapy; Blue triangle denotes treatment response; red circle denotes progressive disease. **b** Waterfall Plot. Depicts percent change from baseline in total sum of target lesion measurements on treatment. Each bar represents a patient, is color-coded to denote best response, and is marked to identify dose level at which they were treated. Source data are provided as a Source Data file. CR complete response, SD stable disease, PD progressive disease.

prior to the administration of nivolumab. Of 21 patients with biopsy specimens evaluable for comparison of PD-L1 expression, TPS was >1% at baseline in 52% of patients (Supplementary Table 5. After IFN-γ induction, 19% revealed an increase in PD-L1 expression compared to baseline, but there was no association with efficacy or dose level (Supplementary 5. CD68+ cells, a marker for tissue-resident macrophages, increased in 60% of specimens after induction, with all but one of the evaluable patients at the two highest dose levels demonstrating an increase after IFN-γ induction (Supplementary Table 6). The assessment of TILs was not evaluable.

**Peripheral blood flow cytometry**
As pre-specified we performed exploratory immune phenotyping of peripheral blood by multi-parametric flow cytometry to quantify subsets of monocytes and T cells, as well as their expression of various biomarkers. Across all four cohorts, while the frequency of classical monocytes remained relatively unchanged throughout the course of treatment (Fig. 2A), there was a statistically significant increase in intermediate monocytes in peripheral blood after IFN-γ induction therapy (C1D1, $p = 0.002$; Fig. 2B) and numbers of non-classical monocytes significantly declined by C2D15 ($p = 0.016$; Fig. 2C). The expression of HLA-DR, which is known to be upregulated by IFN-γ therapy[12], was significantly elevated on the surface of the total monocyte population from pre-induction levels to C1D1 and C2D15 in nearly all patients across all four cohorts ($p = 0.00005$ and 0.004, respectively; Fig. 2D). The increased HLA-DR expression was consistent across all monocyte subpopulations (classical, intermediate, and non-classical; data not shown). PD-L1 expression was significantly increased from baseline levels on intermediate monocytes at C1D1 and C2D15 ($p = 0.01$) and on non-classical monocytes from baseline to C2D15 ($p = 0.004$), and these effects were more prominent in the higher IFN-γ dose (75 and 100 mcg/m²) cohorts (Fig. 2E–G). In addition, a significant correlation was noted between higher levels of PD-L1 expression on intermediate monocytes at C2D15 and shorter duration of treatment ($p = 0.03$, Hazard Ratio = 2.04; Fig. 2H).

Several treatment-related changes were also observed in the peripheral T-cell populations. First, patients with longer duration of therapy tended to have higher numbers of CD4+ T-cells in peripheral blood prior to the start of therapy, whereas those with the lowest frequencies of CD4+ T cells at baseline had shorter duration of treatment ($p = 0.024$, Hazard Ratio = 0.58; Fig. 3A). Also, PD-1 expression levels on both CD4+ and CD8+ T cells declined significantly after the start of nivolumab treatment (C2D15 and C8D1), as compared to levels prior to nivolumab (pre-induction and C1D1; Fig. 3B, C). Decreased expression of PD-1 on T cells after ICB has been reported[13], but the reduction here was significantly more pronounced in patients treated with higher doses of IFN-γ (Fig. 3B, C; Kruskall–Wallis test from pre-induction to C2D15, $p = 0.0065$ for CD4+ T and $p = 0.0059$ for CD8+ T). Finally, for patients that reached C8D1, we observed increased expression of the activation marker CD69 over the course of nivolumab treatment (C2D15 to C8D1) on naïve CD4+ T cells (median increase from 7.63 to 22.5% of cells expressing CD69, $p = 0.016$; Fig. 3D), effector CD4+ T cells (4.11 to 31.0%, $p = 0.031$; Fig. 3E), and naïve CD8+ T cells (10.4 to 31.4%, $p = 0.016$; Fig. 3F).

**Cytokine analysis**
In an exploratory post-hoc analysis performed to assess the relative paucity of irAEs, we noted a statistically significant increase in plasma concentrations of six chemokines from baseline compared to both C1D1 and C2D15. These included known IFN-γ inducible chemokines CXCL9, CXCL10, and CXCL11 (Fig. 4A, $p < 0.005$). The median concentration of these chemokines increased in response to IFN-γ administration and remained elevated with the addition of nivolumab. When the groups were separated by the response (PD vs SD/CR), the increase was only seen in the PD group (Fig. 4B, $p < 0.05$), suggesting the effect was less pronounced in the patients that experienced at least SD. There was no statistically significant difference in the groups by dose cohort or disease sub-type (not shown). Additional chemokines and cytokines that changed from baseline to C1D1 and/or C2D15 are presented in Supplementary Figure 3. We observed increased concentrations of CCL23, CXCL13, and CX3CL1, while levels of CCL2, CCL24, CCL26, GM-CSF, IL-2, and IL-4 were found to decrease after initiation of treatment. Increased

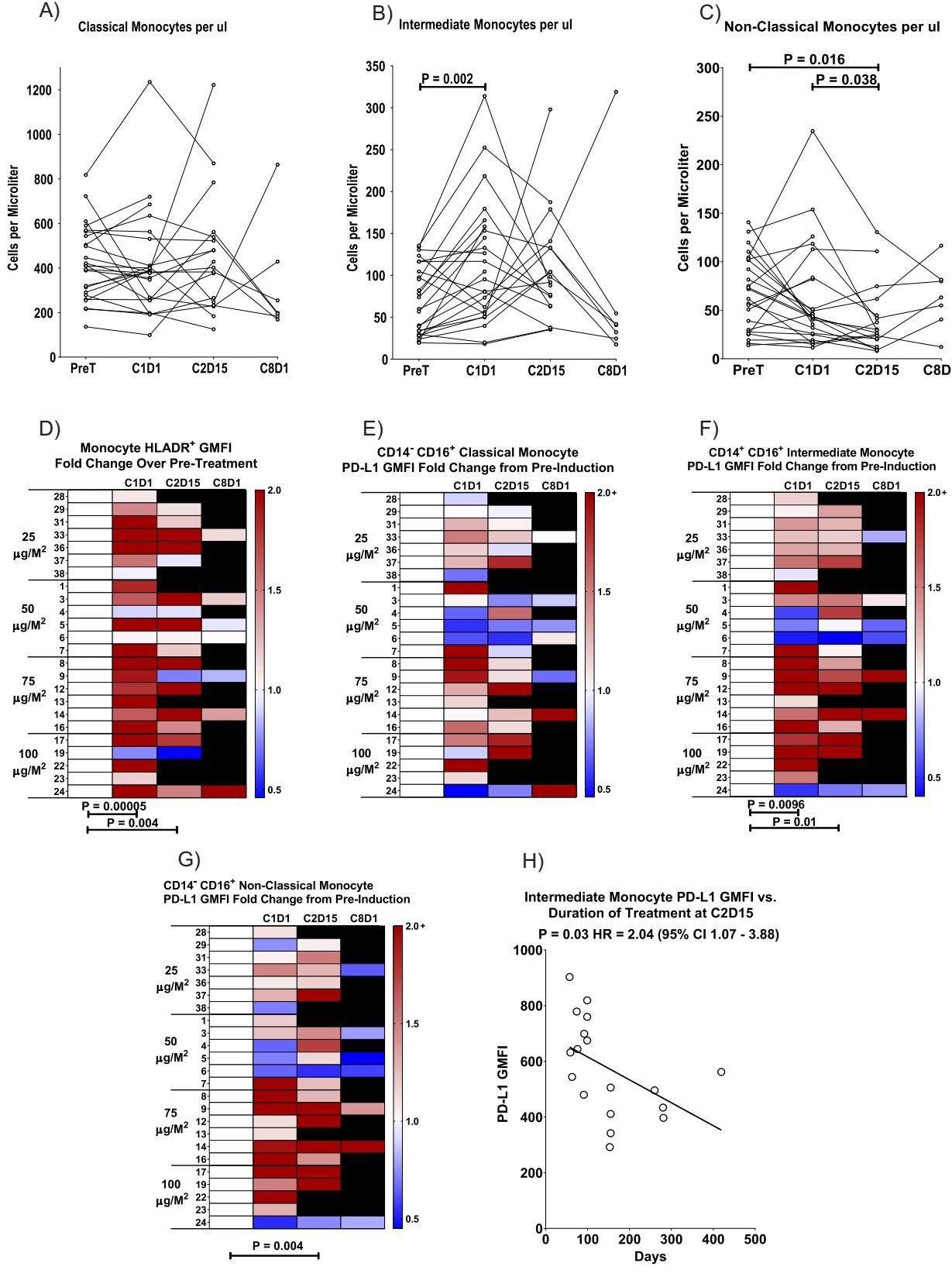

plasma concentrations of IFN-γ were not detected, presumably due to its short half-life and the timing of sample collection.

## Discussion

IFN-γ signaling plays an integral role in regulating immune activation and senescence in the TME, induces the expression of PD-L1 and is an essential component of an effective anti-tumor response with PD-1 pathway targeted agents[14,15]. However, IFN-γ harbors both pro and anti-inflammatory properties and may also be a source of adaptive resistance[10]. Although exogenous administration of IFN-γ to optimize immune activity in combination with anti-PD-1 targeted therapy may be a rational approach, IFN-γ is short-acting, rapidly taken up by

**Fig. 2 | Impacts of IFN-γ and PD-1 blockade on monocytes.** Counts of (**A**) classical (CD14⁺ CD16⁻), (**B**) intermediate (CD14⁺ CD16⁺), and (**C**) non-classical (CD14⁻ CD16⁺) monocytes in peripheral blood were computed by multiplying the Absolute Monocyte Count from a Complete Blood Cell Count by the percentage of in the indicated flow cytometry blood monocyte gate. Each circle represents the measurement on a single patient sample at the indicated time point (PreT pretreatment, C1D1 cycle 1 day 1, C2D15, and C8D1). Samples collected from the same patient are connected by lines, and statistical significance (bars at top) was computed with an exact two-sided Wilcoxon matched-pairs sign-rank test. **D** HLA-DR expression on total monocytes is shown as a function of the indicated time points for each patient. All data are normalized to the expression level (geometric mean fluorescence intensity; GMFI) in the pre-treatment sample (white = 1.0) and shown as a heat map of fold increase (red tones) or decrease (blue tones) from that baseline, as in the scale at the right of each panel. Each row shows the values over time for an individual patient, as numbered at the left, and black indicates no sample for that patient at the indicated time point. Statistical differences between samples assessed at different time points (designated by bars at the bottom) were calculated with an exact two-sided Wilcoxon matched-pairs sign-rank test. Changes in PD-L1 expression during treatment for (**E**) classical monocytes, (**F**) intermediate monocytes, and (**G**) non-classical monocytes are shown in heat map format as a function of the indicated time points for each patient, as numbered at the left. Data are normalized to the pre-treatment sample and shown in heat map format as in (**D**). Statistically significant differences (bottom) were calculated with an exact two-sided Wilcoxon matched-pairs sign-rank test. **H** Correlation of PD-L1 expression level (GMFI) on CD14⁺ CD16⁺ intermediate monocytes at C2D15 to duration of treatment. Each circle represents the data from one patient and the statistical results were computed with a Cox proportional hazard regression test. The line is a least squares fit to the data that is provided for visual purposes. Source data are provided as a Source Data file.

tissues, and its multitude of systemic effects are sensitive to administration dose and schedule.

Overall, the combination was well tolerated. An increased burden of flu-like symptoms was present in most patients, slightly worse at higher doses of IFN-γ, but were generally grade 1-2 and tolerable with supportive care. We did see an unexpected number of patients develop new ascites or pleural effusions, but it is unclear whether this is attributable to IFN-γ. Prior trials of single-agent IFN-γ have not previously reported such a concern. The majority of patients who developed new fluid collections had cancers associated with ascites (i.e., ovarian cancer), or had baseline pleural or peritoneal disease that may have foreshadowed this possibility. One patient developed a pleural effusion after IFN-γ induction without baseline pleural disease that improved upon subsequent anti-cancer therapy without positive fluid cytology. Further trials that involve IFN-γ should consider incorporating planned prospective fluid analysis in patients that develop new ascites or effusions.

Exploratory analyses of efficacy in this trial were modest, with one patient with metastatic TNBC and no prior immunotherapy exposure achieving a durable CR. Although two patients achieved stable disease despite progression on prior ICB, the sample size is too small to make any conclusions. One clear limitation of this trial is the heterogeneous population, with multiple diagnoses that have subsequently demonstrated poorly immunogenic tumors or modest single agent benefit from PD-1 targeted drugs, thus potentially hampering any benefit from IFN-γ. Additionally, the four dose cohorts are small, limiting efforts to assess differences in IFN-γ dose ranges, although more patients with clinical benefit were treated in the two lower dosed cohorts. Based on a combination of increased low-grade AEs and DLTs starting at 75 mcg/m², the longer duration of treatment at the lower dose cohorts, and the increased expression of PD-L1 on peripheral monocytes at the higher dose levels which associated with shorter therapy duration, 50 mcg/m² was chosen as the RP2D. In light of evidence of the anti-immunogenic effects of chronic IFN-γ stimulation[16,17], any future study of this drug in cancer patients should start at doses of 50 mcg/m² or less, with consideration of less frequent dosing or alternative schedules.

Notably, in an exploratory post-hoc observation, there appeared to be a paucity of irAEs amongst patients in this study. The relative incidence of grade ≥ 3 irAE development with nivolumab is about 10-15%. With 26 treated patients, we would have predicted 3-4 events, and yet only one patient developed any manifestations attributable as an irAE, and this only after discontinuing IFN-γ. Several factors may have contributed to this, for example 46.2% patients had received prior ICB without any irAEs and thus may have been less likely to develop an irAE. Additionally, 16 of 23 patients evaluable for efficacy (69.6%) progressed within the first 100 days of therapy, and simply may not have been exposed long enough to nivolumab. It is also possible this was merely by chance. However, it is reasonable to speculate whether the IFN-γ may have altered the immune milieu in such a way to restrain

irAE development. Support for this hypothesis exists with the results of the post-hoc exploratory cytokine analysis, which showed a statistically significant increase in CXCL9, CXCL10, and CXCL11. These three chemokines are known to be directly induced by IFN-γ signaling and bind to a common receptor (CXCR3). Primarily, they are responsible for regulating immune cell migration, differentiation, and activation and thus are important regulators of immune responses in the TME; however, they have also been implicated in promoting tumor proliferation and metastases[18]. A previous retrospective study assessing irAEs in ICB-treated patients found that irAE development was associated with a lower baseline level of CXCL9, CXCL10, and CXCL11 and with a greater increase in these chemokines upon treatment[19]. We postulate that IFN-γ induction may have raised the systemic levels of these key chemokines above a threshold, impacting T-cell activation and diminishing the likelihood of irAE development with the addition of nivolumab. Continued administration of IFN-γ may have maintained these high chemokine levels to prevent autoimmune activation. Physiologically, chronic IFN-γ administration and exposure may induce widespread terminal T-cell exhaustion and PD-L1 independent signaling, essentially rendering T-cells non-functional[17]. Potentially, the IFN-γ may have also attenuated the benefits of nivolumab due to prolonged exposure. There was a statistically significant increase in the same three chemokines in the patients who had progressive disease compared to those with SD or better, further suggesting the chemokine increase may have hampered response. Further prospective research is warranted to evaluate the association between these chemokines and irAE development, as well as whether there could be a role for IFN-γ administration in the prevention or treatment of irAEs.

A number of notable findings were revealed in pre-planned exploratory PBMC flow cytometry studies and biopsy tissue assessments. IFN-γ induction significantly increased the frequency of intermediate monocytes (CD14⁺ CD16⁺) in the peripheral circulation, but these, along with non-classical monocytes (CD14^{low/-} CD16⁺), tended to decline after nivolumab addition. Intermediate monocytes increase in the peripheral blood of patients with infection or inflammatory diseases and have high antigen presentation capacity, whereas non-classical monocytes are more highly differentiated and patrol vascular endothelial surfaces to survey for signs of inflammation before transmigrating into tissues[20–23]. Whether the non-classical monocytes left the circulation to enter the tissues remains unclear, but more CD68⁺ cells (a macrophage marker) were observed in the TME after treatment, and this was accentuated at higher doses of IFN-γ. The upregulation of HLA-DR (MHC Class II) and PD-L1 on all monocyte subpopulations after IFN-γ administration is expected, and indicative that treatment was on-target[12,24]. On the other hand, although IFN-γ administration did not increase PD-L1 expression within the TME, we observed a dose-dependent increase of PD-L1 expression on intermediate and non-classical monocytes, which would be expected to effectively suppress activation of PD-1⁺ T cells[24]. Although nivolumab

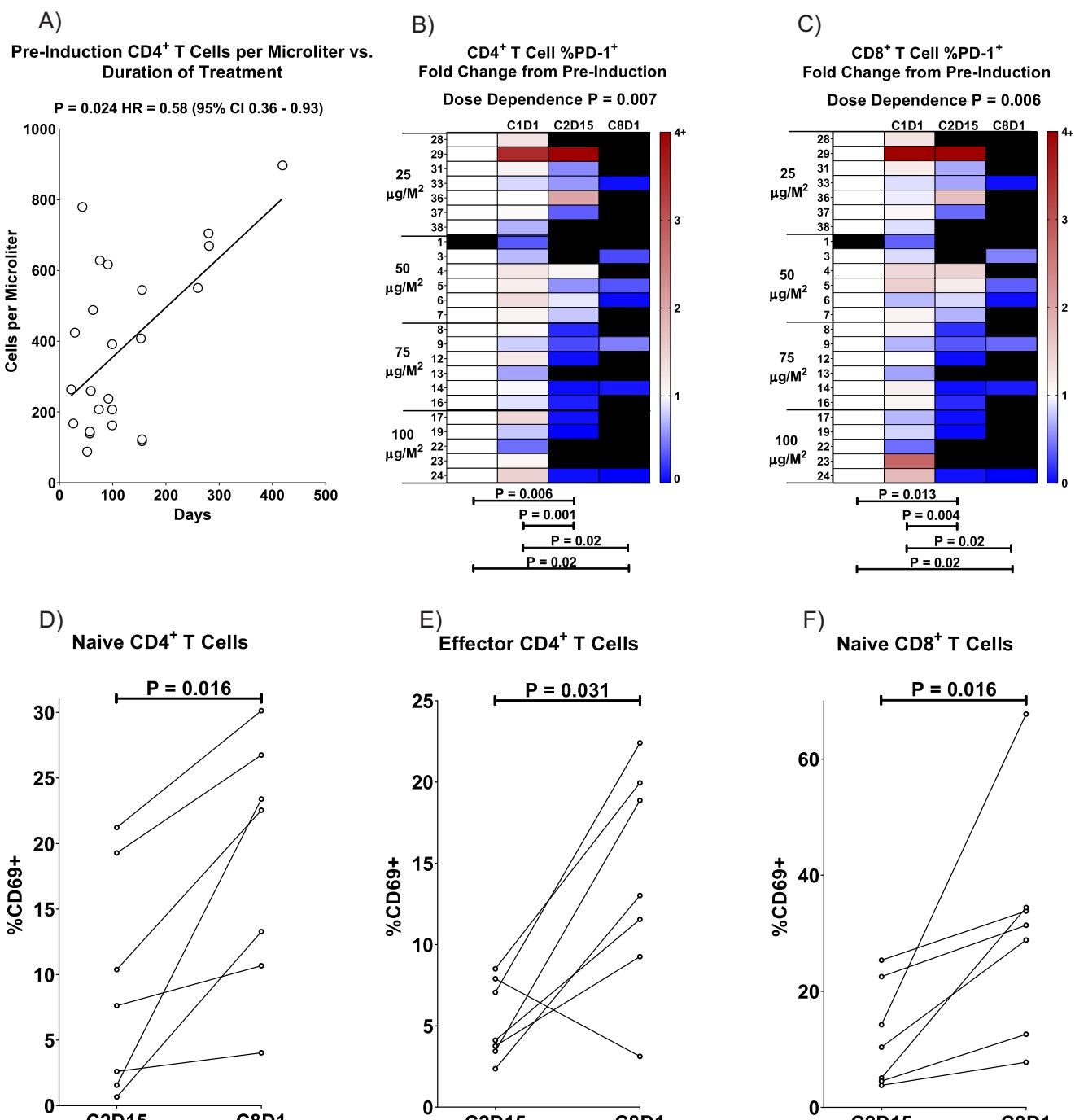

**Fig. 3 | Impacts of IFN-γ and PD-1 blockade on T cells. A** Correlation of pre-treatment (pre-induction) CD4+ T cell counts in blood to duration of treatment. CD4+ T cell counts were computed by multiplying the Absolute Lymphocyte Count from a Complete Blood Cell Count by the percentage of blood CD4+ T cells in the flow cytometry lymphocyte gate. Each circle represents the data from one patient and the statistical results were computed with a Cox proportional hazard regression test. The line is a least squares fit to the data that is provided for visual purposes. **B** Change in percent of CD4+ T cells expressing PD-1 at time points during therapy, as compared to the levels in the pre-treatment samples. Peripheral blood lymphocytes were collected and stained as described in Methods, and surface expression of PD-1 was assayed by primary staining with Nivolumab followed by anti-human IgG4 PE secondary antibody. All data are normalized to the pre-treatment sample (white) and shown as a heat map of fold increase (red tones) or decrease (blue tones) from that baseline, as in the scale at the right of each panel. Each row shows the values over time for an individual patient, as numbered at the left, and black indicates no sample for that patient at the indicated time point. Statistical differences between samples assessed at different time points (designated by bars at the bottom) were calculated with an exact two-sided Wilcoxon matched-pairs sign-rank test. **C** Same as panel B, but for CD8+ T cells. Changes in the percentage of T cell subsets expressing CD69 activation marker over the course of nivolumab therapy from C2D15 to C8D1 are shown on (**D**) naïve CD4+, (**E**) effector CD4+, and (**F**) naïve CD8+ T cells with lines connecting values for individual patients. Statistical differences between time points were calculated with an exact two-sided Wilcoxon matched-pairs sign-rank test. Source data are provided as a Source Data file.

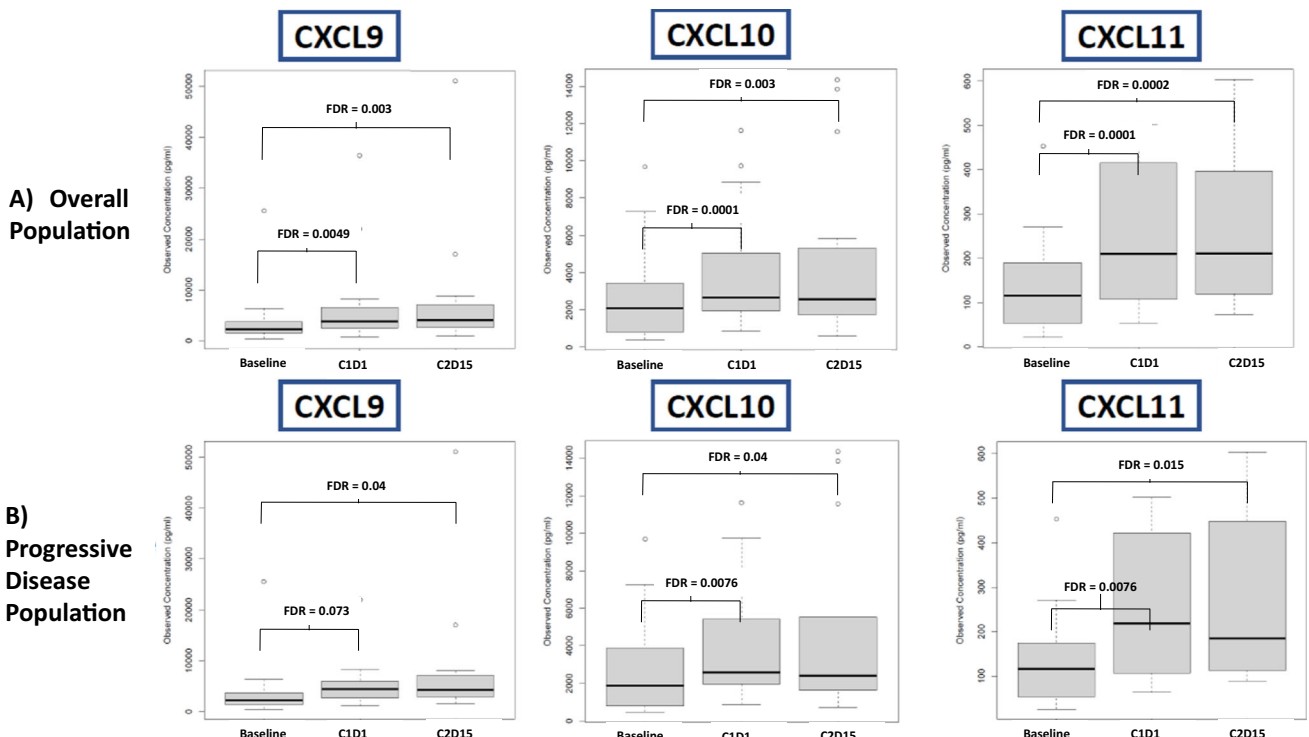

**Fig. 4 | Changes in IFNγ-associated chemokines after induction and addition of nivolumab.** Box plots displaying distribution of chemokine concentrations at three time points on study: baseline, C1D1, C2D15. **A** Distribution of select chemokines across the overall population. Data from $n = 25$ patients were used for the baseline vs. C1D1 comparison and $n = 20$ patients were used for the baseline vs. C2D15 comparison for all 3 markers. **B** Distribution of select chemokines amongst the patients with primary progressive disease. Data from $n = 17$ patients were used for the baseline vs. C1D1 comparison and $n = 14$ patients were used for the baseline vs. C2D15 comparison for all 3 markers. The two-sided Wilcoxon test was used to compare chemokine levels between time points. To account for multiple hypotheses, the Benjamini-Hochberg false discovery rate (FDR) was computed, separately, for each of the different slices per comparison (pair of time-points), and the bracketed differences also had FDR ≤ 0.05. Box plots were plotted using default settings in the R language. In each boxplot, the bold line in the center of the box indicates the median; and the lower and upper hinges represent, respectively, the first and third quartiles. The whiskers are computed based on 1.5 times the inter-quartile range (IQR). They extend to the most extreme data point which is no more than 1.5 times the IQR away from the box. If no points exceed this distance, then the whiskers are the minimum and maximum values. If there are points beyond that distance, then the most extreme point that does not exceed this distance is the whisker. Any data points shown beyond the whiskers are considered outliers. Source data are provided as a Source Data file. IFN-γ interferon gamma, C1D1 cycle 1 day 1, C2D15 cycle 2 day 15.

would be expected to overcome this, patients with the highest levels of PD-L1 on circulating intermediate monocytes had the lowest duration of treatment, suggesting that induction of PD-L1 on these cells by IFN-γ may have hindered efficacy.

Additionally, patients with the lowest frequency of circulating CD4⁺ T cells prior to treatment had the shortest duration of therapy. This observation has also been noted with single agent PD-1 blockade[25,26], since effective PD-1 blockade requires adequate numbers of available exhausted T-cells to be re-invigorated to mount an effective anti-tumor response. In addition, there was a dose-dependent decline in PD-1 expression on CD4⁺ and CD8⁺ T-cells after the addition of nivolumab. Reduced surface expression of PD-1 on T-cells after treatment with blocking antibodies has been previously reported[13], but higher doses of IFN-γ significantly enhanced this effect. It is unclear if the enhanced loss of PD-1 expression on peripheral T-cells is due to augmented endocytosis of PD-1 upon binding of nivolumab or increased reversal of exhaustion, cell death, or tissue migration of PD-1⁺ T-cells.

There were several limitations to this trial that may have impacted the results. The heterogeneous population and small cohort sizes limits comparisons of efficacy and correlatives and dose comparisons. Future trials of larger cohorts in specific cancers, such as RCC, TNBC, or esophagogastric carcinoma, might allow for better assessment of efficacy and a better understanding of the impacts of IFN-γ at the TME in these tumors. The choice of subcutaneous every other day

administration may not have been optimal and may have limited the impact of IFN-γ on the TME and ultimately on efficacy. Lastly, advances and standardization in PD-L1 testing that were not available when this study was designed and conducted may have affected the analytic yield.

In summary, the combination of IFN-γ and nivolumab in previously treated advanced solid tumor patients was safe and based on clinical and correlative findings we recommend 50 mcg/m² as the RP2D. A post-hoc exploratory efficacy analysis showed modest benefit, with 1 CR and a DCR of 26.1%. No patients experienced a grade 3 irAE and post-hoc exploratory cytokine analysis suggests that this lack of irAEs may have been afforded by an IFN-γ-induced increase in the associated chemokines CXCL9, CXCL10, and CXCL11. Further prospective study of the role of IFN-γ in the development of irAEs is warranted.

## Methods
### Study design
The study was conducted in accordance with the principles of Good Clinical Practice and the Declaration of Helsinki and was approved by the Fox Chase Cancer Center (FCCC) Institutional Review Board. This was a single-center, investigator-initiated phase I trial. The study was pre-registered at clinicaltrials.gov and first posted November 25, 2015 (NCT02614456;). Full protocol available in supplementary materials. Patients were accrued to cohorts of six in a modified 3 + 3

design, optimizing patient accrual during a time when access to ICB was more challenging for many tumor types. Four dose finding cohorts of IFN-γ were conducted: 25, 50, 75, and 100 mcg/m$^2$, with a fixed dose initially of 3 mg/kg nivolumab (changed to flat dose of 240 mg with amendment when FDA label changed). Dosing was based on 50 mcg/m$^2$ being the approved dose with 25 mcg/m$^2$ chosen for dose titration. IFN-γ was self-administered subcutaneously every other day (QOD). All patients were started on IFN-γ alone for a one-week induction phase prior to adding nivolumab, then continued combination therapy for up to three months. The inclusion of the induction period was intended to inflame the TME prior to starting ICB, to manage mild IFN-γ-associated AEs, and to facilitate correlative assessments. Patients clinically benefitting after three months would discontinue IFN-γ and remain on nivolumab every three weeks for up to one year. Treatment beyond first progression was allowed if a patient was deemed by the treating investigator to be clinically benefitting.

Patients in each cohort were assessed for dose-limiting toxicities (DLTs) as specified in the protocol (provided in the supplementary material) over the first six weeks of the combination phase, not including the one-week IFN-γ induction since that time period would not have reflected DLTs from the combination. If two or more DLTs occurred in any cohort, that dose of IFN-γ would be deemed intolerable and the prior completed dose would be considered the maximum tolerated dose (MTD). Patients that were not evaluable for DLT because they went off trial for disease progression or withdraw of consent without receiving 75% of IFN-γ doses were replaced, but were included in overall safety analysis. Based on a complete assessment of toxicity, as well as correlative factors, the study team had authority to denote the MTD as the recommended phase two dose (RP2D), or if warranted for safety and/or scientific reasons, a lower dose could also be chosen as the RP2D.

Patients were seen every two weeks once they entered the combination phase, then every three weeks during the single agent phase. RECIST version 1.1 was used to assess objective tumor response, with assessments performed after the combination phase and then every 9 weeks[27].

All patients provided written informed consent and were not compensated for study participation. The first patient was enrolled 12/29/2015 and the last patients was enrolled 2/12/2018. A steering committee consisting of select study investigators and staff met after each dose cohort to review safety data and decide whether to proceed to the next dose, and the study conduct was also reviewed regularly by an independent FCCC data safety monitoring board. The study was conducted at and sponsored by Fox Chase Cancer Center and all data was evaluated independently. Funding for study conduct and IFN-γ administration was provided by Horizon Pharma, LLC. The original design included plans for expansion cohorts to assess preliminary efficacy as measured by response rate in specific cancers of interest; however, the funding organization elected to halt further trial conduct after the Phase I portion. The original plan can be found in the full protocol in the supplement, but no changes to analysis or design of the phase I portion were made.

## Study population

Eligible patients were ≥ 18 years of age, and could have any metastatic solid tumor type where there was demonstrated evidence of potential efficacy to anti-PD-1 pathway targeted therapy at the time of study design. All genders were eligible and patients self-reported. All patients needed to have received at least one prior systemic anti-cancer therapy; prior ICB was allowed, as long as the reason for discontinuation was not an irAE. A site of disease amenable to a fresh biopsy was required for inclusion. A two-week washout period for previous systemic therapy was required, and patients could not have a prior history or concomitant autoimmune condition.

## Study objectives

The primary endpoint of the study was the safety of the combination and to establish a RP2D. A final RP2D was to be established by the study steering committee at the conclusion of all accrued dose cohorts. Initial secondary efficacy endpoints included PFS, OS (median and at 1 year), and ORR for patients in the planned expansion cohorts of esophagogastric carcinoma and in patients with metastatic cancers refractory to prior PD-1 targeted treatment. We also planned to investigate the relationship between PD-L1 expression on tumor cells and on immune cells in the tumor microenvironment before and after treatment initiation. The expansion cohorts were not pursued due to perceived limited efficacy by the funding body, and a decision was made to perform exploratory post-hoc analyses to evaluate efficacy in the dose escalation patients. Pre-planned exploratory objectives included investigating changes in PD-L1 expression in tumor biopsy samples and in peripheral blood in relation to response, as well as the effect of IFN-γ administration on changes in known IFN-γ markers. Post-hoc, we elected to perform an exploratory cytokine analysis from plasma at various time points to associate with incidence of irAEs.

## Pre-specified exploratory correlative assessments

All patients were required to undergo a baseline tumor biopsy within 28 days prior to starting trial treatment, and then underwent a second tumor biopsy of the same site after the IFN-γ induction phase but before receiving nivolumab. Tumor specimens were to be evaluated by immunohistochemistry for PD-L1 expression and immune cell infiltration as pre-planned exploratory analysis. Biopsy samples underwent routine pathology review to determine cellularity and tumor presence via hematoxylin and eosin stain. PD-L1 expression was determined via immunohistochemistry (IHC) utilizing VENTANA PD-L1 (SP263) and calculated using the tumor proportion score (TPS)[28]. IHC for CD68 was performed according to local protocols on an automated immunostainer (Ventana Benchmark Ultra: Ventana Medical Systems, Tucson, Arizona). Mononuclear intratumoral and stromal tumor infiltrating lymphocytes (TILs) were scored for each sample[28–30]. All samples were reviewed by a single pathologist. In a protocol-specified exploratory analysis, peripheral blood was drawn for correlatives (1) at baseline, (2) after IFN-γ induction but prior to starting nivolumab (cycle 1 day 1; C1D1), (3) after three doses of nivolumab (C2D15), and (4) during single-agent phase for patients still on trial (C8D1). Peripheral blood was separated into plasma and peripheral blood mononuclear cells (PBMCs). PBMCs were analyzed the same day by the FCCC Immune Monitoring Facility using 12-color multiparametric flow cytometry employing the antibody staining panel shown in Supplementary Table 4 to quantify biomarkers on T, NK, and myeloid cells utilizing leukocyte sub-gating strategies, as shown in supplementary figures 4 and 5 and previously described[25,31]. Fluorophore-conjugated antibodies (volumes shown in Supplementary Table 4) were added to 1 million PBMCs in 100 μL staining buffer (Hanks's Balanced Salt Solution + 1% heat-inactivated fetal bovine serum and 0.09% sodium azide), incubated on ice × 20 min, and washed twice with staining buffer before analysis. Surface PD-1 was measured by primary staining of those samples with unlabeled nivolumab (1 μg in 100 μl wash buffer on ice × 20 min), followed by two washes, secondary staining with fluorophore-conjugated anti-human IgG4 (10 μL in 100 μL staining buffer on ice × 20 min), and two washes on ice, as previously described[13,25]. Flow cytometry data were acquired on a BD Aria II sorter and processed using FlowJo software (BD; version 10.7). DNA was extracted and prepared for future genotyping.

## Post-hoc correlative analyses

In a post-hoc analysis prompted by clinical observations during the trial, cytokines in plasma were analyzed by the FCCC High Throughput Screening Facility utilizing the Human Chemokine Panel 40-plex (Bio-Plex Pro, Bio-Rad cat # 171AK99MR2) following manufacturer's protocols and plates were read on a Bio-Plex 100/200 (Bio-Rad

Laboratories, Hercules CA). Plasma underwent cryopreservation at −80 °C via standard operating procedures and was later used for batch cytokine analysis. The instrument was driven with Bio-Plex Manager (version 6.1.0727) software and data was analyzed and exported utilizing Bio-Plex Data Pro software (version 1.2.03).

## Statistical analysis

PFS and OS in all evaluable patients were calculated using Kaplan-Meier curves. PFS was the interval to progression or death in those evaluable for efficacy. Duration of treatment was defined as the date from first dose of IFN-γ to progression, death, or completion of all treatments. The ORR was calculated using RECIST v 1.1. Planned exploratory correlative analysis of the difference in immune parameters at various treatment time points was done using a Wilcoxon test where consecutive samples from the same patient constituted a pair. Correlations to duration of treatment were performed using a Cox proportional hazards regression and tests for dose dependence were done using a Kruskal–Wallis test. *P*-values less than 0.033 were considered to be statistically significant. Calculations were done with Matlab R2016b Statistics and Machine Learning Toolbox.

For the post-hoc cytokine analysis, a two-sided Wilcoxon test was used to compare cytokine levels and in order to account for multiple hypotheses, the Benjamini–Hochberg false discovery rate (FDR) was computed, separately, for each comparison performed (baseline vs. C1D1, baseline vsC1D15, C1D1 vs C1D15). Cytokines with FDR corrected *p* value ≤ 0.05 were considered to be statistically significant. Computations were made using the R language[32].

## Reporting summary

Further information on research design is available in the Nature Portfolio Reporting Summary linked to this article.

# Data availability

Source data are provided with this paper. The minimum dataset necessary to interpret, verify and extend this research has been provided within the source data file wherever applicable. Per ICMJE guidelines de-identified participant data has been provided within the source data file. The study protocol has been provided as supplementary material. Source data are provided with this paper.

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

## Acknowledgements

The authors thank Irina Shchaveleva, Chun Zhou, and Judy Fang from the FCCC Immune Monitoring/Cell Sorting Facility for technical support. Jeffrey Sherman, Jeffrey Nieves, and Amy Grahn for support from Horizon Pharma. The NCI Cancer Center Support Grant (CA06927; FCCC) for support of core facilities utilized (Immune Monitoring/Cell Sorting, High Throughput Screening, and Biostatistics & Bioinformatics Facilities) and philanthropic funds from the FCCC In Vino Vita fundraiser, FCCC Bucks County and Mainline Boards of Associates, C.W.A. Local 13000 Jim Willer Golf Tournament, and the FCCC Dragon Boat Team. Employees of Horizon Therapeutics did not contribute to or influence study design, data collection, or data analysis. Full support for clinical trial provided by Horizon Therapeutics, PLC.

## Author contributions

M.Z. is the corresponding author, conceptualized and designed the study, was the principal investigator of the clinical trial, coordinated data analysis and wrote and edited the manuscript. E.R.P. provided oversight for trial design and conduct, participated in data analysis and provided editorial support. K.S.C. coordinated primary correlative analysis plans, oversaw the flow cytometry analysis, and provided writing and editorial support. A.W.F. performed the flow cytometry work and analysis and provided writing and editorial support. K.C., T.M., J.O. coordinated trial activities and helped with data collection and analysis. RK oversaw trial conduct, coordinated specimen collection and analysis, provided writing and editorial support. H.B., C.S.D., E.D., D.M.G., A.J., L.M., E.O. participated in patient accrual and provided editorial support. K.D., K.R. provided statistical analysis for clinical and correlative data and provided writing and editorial support. R.K.A. oversaw collection and processing of all correlative specimens and provided data analysis. E.A.D. prepared and analyzed all pathology specimens. E.C., M.E. performed multiplex and cytokine analyses and provided data analysis support and editorial support. All authors had an opportunity to review and approve the final submitted manuscript and

## Competing interests

M.Z.: Institutionally directed research funding from Horizon Therapeutics and Bristol-Myers Squibb; Advisory Board Honorarium from Horizon Therapeutics. K.S.C.: Advisory Board Honorarium from Horizon Therapeutics, institutionally-directed research funding from Bristol-Myers Squibb. E.R.P.: Advisory Board Honorarium from Horizon Therapeutics. H.B.: research support (clinical trials) from BMS, Lilly, Amgen; Advisory Board/Consultant for BMS, Lilly, Genentech, Pfizer, Merck, EMD-Serono, Boehringer Ingelheim, Astra Zeneca, Novartis, Genmab, Regeneron, BioNTech, Amgen, Axiom, PharmaMar, Takeda, Mirati, Daiichi, Guardant, Natera, Oncocyte, Beigene, iTEO, Jazz, Janssen, Da Volterra, Puma, BerGenBio, Bayer, Iobiotech; Data and Safety Monitoring Board for University of Pennsylvania (CAR T Program), Takeda, Incyte, Novartis, Springworks; employment at Fox Chase Cancer Center; Scientific Advisory Board for Sonnetbio (stock options), Inspirna (formerly Rgenix, stock options), Nuclei (stock options); honoraria from Amgen, Pfizer, Daiichi, Regeneron; travel support from Amgen, BMS, Merck, Lilly, EMD-Serono, Genentech, Regeneron. The remaining authors declare no competing interests.
