## [Peer Review File · Nature Communications]

REVIEWER COMMENTS

Reviewer #1 (Remarks to the Author):

Dr. Zibelman and colleagues conducted a single center investigator-initiated phase I trial combining anti PD1 nivolumab with interferon gamma (INF-gamma). The trial included a total of 26 patients with various advanced solid tumor malignancies either immunotherapy pre-exposed or naïve (the majority). All patients did receive at least 1 prior line of therapy. Four dose levels of INF-gamma self-administered subcutaneously every other day were evaluated in this 3+3 modified escalation design. The study comprised of a lead-in period of INF-gamma followed by a combination phase (INF-gamma with nivolumab) and ultimately maintenance of nivolumab for up to 1 year. Tissue and blood based biomarker analyses were performed. The primary objectives were safety/tolerability and dose finding. The authors defined the combination of INF-gamma and nivolumab tolerable with no grade 3 or higher immune-related adverse events and low grade flu-like symptoms the most common toxicities. However, fluids accumulation (ascites and pleural effusion) were reported in several patients. The MTD of INF-gamma in combination with nivolumab was 75mcg/m² and the RP2D was defined at 50 mcg/m². In terms of efficacy the authors reported one complete response (TNBC patient naïve to immunotherapy), zero partial responses, 5 stable disease, while the majority of the patients did progress. Correlative analyses were incorporated.

Overall, although interferon gamma is a critical cytokine able to influence anti-tumor immunity by suppressing tumor growth via disparate mechanisms, its role as anticancer therapeutic agent has been limited and disappointing. The rationale to attempt a combination study with antiPD1 is well outlined however the results of this small phase I study are not a clear home run both in terms of safety and signal of efficacy. Below I have outlined my comments on specific aspects of the study report and results.

Abstract: the abstract is lacking safety details and DLTs

Safety:

- Taken into consideration that, as a phase I study, the safety is the primary objective, I find the presentation of the toxicity profile of this study very limited and unsatisfactory. I suggest major revisions of Table 2 to include adverse events at least possibly related to both agents divided by dose level and grading (examples G1-2; G \geq 3). Table 2 should also highlight the DLTs seen in this study. This changes will add comprehensive and meaningful data. Finally should the toxicities be organized by organ/system? It is unclear if AST/ALT elevation (together seen in 96%) of cases, is reflective of hepatitis. Why described separately?
- Perhaps as important, a supplementary safety information of the lead-in phase could be added as lead-in is obviously not included in the DLT period.
- The description in the text of the safety is vague and no % is added to the main drug-related adverse events such as hepatitis or the fluids accumulation which to me are relevant given the agents involved.
- DLTs: 2 patients experienced G3 fatigue that were defined DLTs. G3 fatigue events were attributed to both drugs but not defined immune-related. In addition G3 LFTs increased were seen. Could the authors clarify why they believe this was not immune-related?
- I think it would be informative to report data regarding treatment discontinuation (how many patients) and treatment delay (how many patients and cause).
- In the discussion the authors attempt to justify the events of ascites and pleural effusions based on the fact that the patients who experienced such events had a cancer highly associated with ascites. Have the author explored potential INF-gamma related mechanisms of fluid accumulation based on previous literature? Were therapeutic paracentesis and thoracentesis performed and were cytology positive for cancer or were inflammatory findings detected?

Efficacy:

- Figure 1 summarizes the duration of response in 26 patients enrolled in the study. Based on the color code the number of stable disease (light blue) were 6 and not 5 as reported in the manuscript. Could you please clarify this discrepancy?
- The patient with CR had TNBC (labeled as breast cancer in Figure 1 but listed as TNBC in table 1) was immunotherapy naïve. Out of the 6 patients (Figure 1 light blue color code) with SD, 4 were immunotherapy naïve and included mostly immunotherapy sensitive tumor types (RCC, esophageal and endometrial cancers). In addition, I suggest the authors acknowledge that 4/6

patients with SD had a duration of response < 6months.

- Had the patients pre-exposed to immunotherapy with SD as best response in this study, primary or acquired resistance to immunotherapy?
- The efficacy section in the discussion should be revised as most patient who had a benefit were immunotherapy naïve.
- I believe adding waterfall plot to evaluate tumor shrinkage or growth pattern in the patients defined to have DCR would be informative.
- Line 336: "minimal benefit from antiPD1 targeted therapy". I do not fully agree with the authors as most of immunotherapy sensitive tumors were included in the study.

Correlative Analysis:

- I congratulate the authors for collecting tumor and blood for correlative analysis in this phase I trial. I believe the data presented are very informative however conclusions are limited by numbers.
- PDL-1 expression was performed but the assay use is not representative of what currently used for various tumor types. I recognize the authors have listed this matter as limitation. Although PDL1 expression is used in certain tumor type as a biomarker selection, we all acknowledge the limitation of this biomarker.
- The justification of the impact of INF-gamma therapy on paucity of irAEs could be strengthened by a clear description of the events and appropriate explanation of exclusion of irAEs cause.

Additional comments:

- In the study design section authors state that nivolumab was given at 3mg/kg. However, in the protocol schema nivolumab flat dose of 240 mg IV is reported. Could author please clarify such discrepancy?
- Based on what data the authors decided to proceed with nivolumab 240 mg flat dose Q3W in the maintenance portion of eth study instead of 240 Q2W or 480 mg Q4W?
- Study objective section: please clarify that the primary endpoint is safety and tolerability. Currently the wording is "Primary endpoint of RP2D" and there is no mentioning of the safety.

Reviewer #2 (Remarks to the Author):

This article reports on a phase 1 trial administering interferon-gamma systemically together with anti-PD-1 to patients with solid malignancies. It is a truly first-in-human trial based on a testable hypothesis, with reporting of patient-derived samples to analyze the effects of therapy on patients.

Main comments:

Why did it take from 2018 until now to submit this article? The long-term follow-up of patients does not seem to justify the delay in reporting.

Since there is a staggered start of the therapy and the interferon gamma seems to only have been administered for the first three months, it would be useful to include in Figure 1 a schematic of the treatment timeline as otherwise it is only buried in the text of the Methods.

The swimmer's lane plot in Figure 1 should include arrows for any patients who continue to respond at the time of reporting, and symbols depicting the time that a patient achieved the best response on therapy (CD, SD) and the time of subsequent disease progression.

Twenty-one patients underwent tumor biopsies, but the results of their analyses are buried in supplemental tables. They also include very limited analyses, which seems a missed opportunity. Even without more data being generated, it would be of interest to include representative IHC images of changes in PD-L1 and immune cells in a main article figure.

Minor comments:

The Discussion is unnecessarily long, with way too much repetition of the results presented immediately before.

Reviewer #3 (Remarks to the Author):

This is a phase I dose-escalation study for combined IFN-gamma and nivolumab in patients with advanced solid tumor. Totally n=26 for 4 dose levels (25, 50, 75 and 100). There is 1 DLT out of 6 subjects on 75 dose level and 2 DLT out of 5 subjects on 100 dose level, so based on 3+3 design, MTD is dose 75. Based on the longer duration of treatment (why not the duration of response ?) at dose level 50, the RP2D is dose 50, which is reasonable. The statistical approaches used in the correlative data analysis looks good in general. Though, some major comments as below:

1. There are some discrepancy between "study objectives and statistical analysis" section and results section. For example, "secondary endpoints for PFS and OS were calculated using Kaplan-Meier curves" but the curves were not provided. Also, the definition of PFS is not clear. "PFS was the interval to progression or date of last follow up...", which is not correct since PFS includes two events progression or death which occurs earlier. Need to clarify.
2. The time-to-event endpoint "Duration of treatment" is not well defined. It's not clear what is the events for duration of treatment. Whether it means the time from receiving the first treatment to (1) Event No.1 as "finished all the treatments" (2) Event No.2 as "death" (3) or if no event, it censored at "last follow-up time". If so, every subject has an event ? No subject was censored ? This endpoint has been used in figure 2H and 3A with a Cox model, so it needs more details.
3. Similarly, Figure 1 is confusing. First, the title is "Duration of response" but the x-axis is "weeks on treatment". I do not think they mean the same thing. For the duration of the response, it needs to specify when the response starts and when the response ends (e.g., death or disease progression after a response), it's not clear whether the end of each bar means this subject is dead at this time or just the last follow-up and still alive.

RESPONSE TO REVIEWERS' COMMENTS

Please see below point by point response to the reviewer's comments. Reviewer's remarks are denoted in italics, while the author responses can be found following each one in red font.

Reviewer #1:

- Abstract: the abstract is lacking safety details and DLTs:

We have modified the abstract to include more specifics regarding the DLTs and safety on trial, including MTD and RP2D, within the limits of abstract word count.

Safety:

- Taken into consideration that, as a phase I study, the safety is the primary objective, I find the presentation of the toxicity profile of this study very limited and unsatisfactory. I suggest major revisions of Table 2 to include adverse events at least possibly related to both agents divided by dose level and grading (examples G1-2; G≥3). Table 2 should also highlight the DLTs seen in this study. This changes will add comprehensive and meaningful data. Finally should the toxicities be organized by organ/system? It is unclear if AST/ALT elevation (together seen in 96%) of cases, is reflective of hepatitis. Why described separately?

Thank you for the suggestions. We have updated and expanded table 2 to include the reviewer's suggestions. The table now includes all AEs at least possibly related to either drug, sorted by frequency with most common at the top, and breaking it down amongst the entire group and by dose level. An asterisk denotes DLTs.

- Perhaps as important, a supplementary safety information of the lead-in phase could be added as lead-in is obviously not included in the DLT period

A new table summarizing the AEs during IFN induction is now included in the supplemental section as supplemental table S1.

- The description in the text of the safety is vague and no % is added to the main drug-related adverse events such as hepatitis or the fluids accumulation which to me are relevant given the agents involved.

We have updated the text description and have provided additional documentation. In regards to the fluid accumulation, we have updated the language in lines 241-247 to improve clarity, and a new table is provided in the Supplement labelled Table S2 More specific data regarding elevated transaminase levels, which is the preferred CTCAE

terminology, is now included in lines 233-236 and the updated Table 2.

- DLTS: 2 patients experienced G3 fatigue that were defined DLTs. G3 fatigue events were attributed to both drugs but not defined immune-related. In addition G3 LFTs increased were seen. Could the authors clarify why they believe this was not immune-related?

Neither of these patients had any evidence of an immune related adverse event that would have correlated with their fatigue. There was no change in TSH, cortisol level, hemoglobin, or other end organ change that could account for the worsening grade of fatigue. The treating investigators ultimately deemed that that change in status could not be clearly attributed to another cause, and hence was attributed as possibly related to both drugs. This was determined by the investigators given the data available at the time the patient experienced the AE and was adjudicated by the study steering committee. While not standard at the time this study was conducted, most trials now investigating checkpoint inhibitors differentiate clearly diagnosed/defined immune related adverse events when determined, from other AEs at least possibly attributed to drug (if no other clear cause) but not defined as an irAE. Both of these patients ultimately had evidence of disease progression soon after coming off trial and died of their disease. Some clarifications have been made for clarity in the manuscript between lines 229-233.

- I think it would be informative to report data regarding treatment discontinuation (how many patients) and treatment delay (how many patients and cause).

In regards to discontinuation, all patients went off trial for disease progression, except for the two patients already mentioned in lines 223-227 who discontinued for IFN-related AEs. One patient completed therapy with a CR. No patients stopped nivolumab for AEs. Three patients had a dose of nivolumab skipped for AEs day of planned treatment, though none ultimately determined to have an irAE. This was added to the manuscript in lines 225-227.

- In the discussion the authors attempt to justify the events of ascites and pleural effusions based on the fact that the patients who experienced such events had a cancer highly associated with ascites. Have the author explored potential INF-gamma related mechanisms of fluid accumulation based on previous literature? Were therapeutic paracentesis and thoracentesis performed and were cytology positive for cancer or were inflammatory findings detected?

There have been many trials with interferon-gamma in a cancer population and ascites and pleural effusions have not been widely reported. This point is already included in the discussion section in line 329-331. Management of these was as per standard of care per

treating clinician, so documentation of which patients had positive cytology sent was not included as part of the trial. Please see additional supplemental table S2 for more information regarding these events.

Efficacy:

- *Figure 1 summarizes the duration of response in 26 patients enrolled in the study. Based on the color code the number of stable disease (light blue) were 6 and not 5 as reported in the manuscript. Could you please clarify this discrepancy?*

Thank you for noticing this discrepancy, which was not intentional. The correct number of stable disease patients is 5 as stated in the manuscript. This was a color error in the figure and has been updated in the re-submitted revision.

- *The patient with CR had TNBC (labeled as breast cancer in Figure 1 but listed as TNBC in table 1) was immunotherapy naïve. Out of the 6 patients (Figure 1 light blue color code) with SD, 4 were immunotherapy naïve and included mostly immunotherapy sensitive tumor types (RCC, esophageal and endometrial cancers). In addition, I suggest the authors acknowledge that 4/6 patients with SD had a duration of response < 6months.*

All comments acknowledged. We would disagree that endometrial cancer is generally considered an immune-sensitive cancer, however we are in total agreement that efficacy here is minimal. We have modified the discussion section on efficacy to address this (lines 337-344), and I have also added a comment in the results section (lines 263-264) regarding short duration of SD in 3/5 pts. We also added a line about an esophageal patient with clear clinical benefit, but who had a new lesion equating to PD, and a treatment duration exceeding 6 months.

- *Had the patients pre-exposed to immunotherapy with SD as best response in this study, primary or acquired resistance to immunotherapy?*

It's a good question, however that information was not collected as part of the study, and neither patient was on a prior study, so accurate data regarding response to prior treatment is not available.

- *The efficacy section in the discussion should be revised as most patient who had a benefit were immunotherapy naïve.*

As noted above we have modified the discussion section on efficacy to address this point.

- I believe adding waterfall plot to evaluate tumor shrinkage or growth pattern in the patients defined to have DCR would be informative.

Thank you for this suggestion, we think this does add informative visual information to the treatment efficacy. The waterfall plot has been added to the main part of the manuscript as Figure 1b.

- Line 336: "minimal benefit from antiPD1 targeted therapy". I do not fully agree with the authors as most of immunotherapy sensitive tumors were included in the study.

Of the tumor types included in this study, 13/26 patients (50%) had diagnoses without an improved checkpoint inhibitor used as a single agent, which was the case in this study. Of the remaining tumor types, RCC, urothelial and esophagogastric, all had single agent response rates < 25%, which was in immunotherapy naïve patients. However, we have modified the language of this sentence to soften the statement.

Correlative Analysis:

- I congratulate the authors for collecting tumor and blood for correlative analysis in this phase I trial. I believe the data presented are very informative however conclusions are limited by numbers.

Thank you, we agree

- PDL-1 expression was performed but the assay use is not representative of what currently used for various tumor types. I recognize the authors have listed this matter as limitation. Although PDL1 expression is used in certain tumor type as a biomarker selection, we all acknowledge the limitation of this biomarker.

Agree

- The justification of the impact of INF-gamma therapy on paucity of irAEs could be strengthened by a clear description of the events and appropriate explanation of exclusion of irAEs cause.

We have attempted to describe the irAEs in this study, however only one event, which is described in details from lines 250-254, met criteria for an irAE. No other patient had even a change in thyroid function requiring new treatment or management of a low grade rash potentially attributable to nivolumab. As discussed in the discussion section, this is much

less than would have been expected, however there are multiple reasons why this may have occurred separate from the effect of IFN-gamma, and we review these in the discussion. Still, in line with the changes in chemokines described, we believe there is certainly a case to be made that IFN-gamma could have a role here and would require further dedicated study to determine.

Additional comments:

- In the study design section authors state that nivolumab was given at 3mg/kg. However, in the protocol schema nivolumab flat dose of 240 mg IV is reported. Could author please clarify such discrepancy?

Thank you for pointing this out. As stated in the protocol on page 24, the second amendment changed the dosing of nivolumab in line with the change in FDA labelling at that time. The initial patients were treated at the weight-based dosing, and then subsequent patients who accrued during the second amendment and beyond received fixed dosing. We have updated this in the methods section in lines 117-119.

- Based on what data the authors decided to proceed with nivolumab 240 mg flat dose Q3W in the maintenance portion of eth study instead of 240 Q2W or 480 mg Q4W?

Obviously, there is no data to support the exact 3-week dosing. There are however data from McDermott et al. (*J Clin Oncol.* 2015 Jun 20; 33(18): 2013–2020) demonstrating similar efficacy and toxicity at 1 or 10 mg/kg in patients with metastatic RCC. The PKs from this and other studies have demonstrated complete receptor occupancy at very low doses. Since the nivolumab at that time was being purchased for the trial, it was less expensive to dose at the fixed dosing so as not to waste any drug, and based on the data available at the time, the dose and schedule did not seem likely to impact efficacy or toxicity. Thus, once fixed dosing became available, we felt it made the most sense to treat at the updated dose, but not alter the schedule of the entire trial.

- Study objective section: please clarify that the primary endpoint is safety and tolerability. Currently the wording is "Primary endpoint of RP2D" and there is no mentioning of the safety.

Correct. Wording clarified at lines 175-177.

Reviewer #2:

Why did it take from 2018 until now to submit this article? The long-term follow-up of patients does not seem to justify the delay in reporting.

Fair question. All authors wish it was published long ago. The last patient on trial completed treatment in 2019. Cytokine and tumor tissue studies were batched and performed at the completion of all patients on trial. Initially, we planned for further expansion cohorts, but the sponsor withdrew further support. So, at the end of 2019 we began to prepare for submission and then the pandemic hit, which delayed our ability to submit for a variety of reasons as you might imagine. The first version of the manuscript was finally submitted in early 2021, with subsequent resubmissions. Each resubmission required extensive edits after months of review. Throughout the course of revisions, we addressed weaknesses to strengthen the paper for subsequent submission.

Since there is a staggered start of the therapy and the interferon gamma seems to only have been administered for the first three months, it would be useful to include in Figure 1 a schematic of the treatment timeline as otherwise it is only buried in the text of the Methods. The swimmer's lane plot in Figure 1 should include arrows for any patients who continue to respond at the time of reporting, and symbols depicting the time that a patient achieved the best response on therapy (CD, SD) and the time of subsequent disease progression.

Thank you for these suggestions. We have made the changes including adding marks for progression and best response. No patients remain on therapy as the one patient with a CR completed all therapy on the trial. We also incorporated a line at the bottom of the swimmer's plot to serve as a schema pictorializing the periods of treatment (IFN induction, combination, single agent Nivolumab).

Twenty-one patients underwent tumor biopsies, but the results of their analyses are buried in supplemental tables. They also include very limited analyses, which seems a missed opportunity. Even without more data being generated, it would be of interest to include representative IHC images of changes in PD-L1 and immune cells in a main article figure.

We agree, however, owing to the lack of available standardized PD-L1 testing at the time this study was done and poor quality of some biopsy specimens there was not enough useful material to have data worthy of including in this publication. We had hoped this would be a key component of our analysis for this study, but the data were not robust enough to include.

Minor comments:

The Discussion is unnecessarily long, with way too much repetition of the results presented immediately before.

We streamlined the Discussion and limited repetition of describing the results.

Reviewer #3:

This is a phase I dose-escalation study for combined IFN-gamma and nivolumab in patients with advanced solid tumor. Totally n=26 for 4 dose levels (25, 50, 75 and 100). There is 1 DLT out of 6 subjects on 75 dose level and 2 DLT out of 5 subjects on 100 dose level, so based on 3+3 design, MTD is dose 75. Based on the longer duration of treatment (why not the duration of response ?) at dose level 50, the RP2D is dose 50, which is reasonable. The statistical approaches used in the correlative data analysis looks good in general. Though, some major comments as below:

1. There are some discrepancy between "study objectives and statistical analysis" section and results section. For example, "secondary endpoints for PFS and OS were calculated using Kaplan-Meier curves" but the curves were not provided.

We made an initial decision not to include these in the submission given the wide range of tumor types included in the study. It did not seem that instructive to have KM curves for survival. However, we have included these now in the supplement as supplemental figures S1 and S2.

Also, the definition of PFS is not clear. "PFS was the interval to progression or date of last follow up...", which is not correct since PFS includes two events progression or death which occurs earlier. Need to clarify.

Correct, thank you, we updated this definition in lines 177-178 in the Methods section.

2. The time-to-event endpoint "Duration of treatment" is not well defined. It's not clear what is the events for duration of treatment. Whether it means the time from receiving the first treatment to (1) Event No.1 as "finished all the treatments" (2) Event No.2 as "death" (3) or if no event, it censored at "last follow-up time". If so, every subject has an event ? No subject was censored ? This endpoint has been used in figure 2H and 3A with a Cox model, so it needs more details.

We have provided the definition used in the Methods section.

3. Similarly, Figure 1 is confusing. First, the title is "Duration of response" but the x-axis is "weeks on treatment". I do not think they mean the same thing. For the duration of the response, it needs to specify when the response starts and when the response ends (e.g., death or disease progression after a response), it's not clear whether the end of each bar means this subject is dead at this time or just the last follow-up and still alive.

Thank you for these comments. We have updated the title of Figure 1 (now figure 1a) to more precisely reflect the data being shown, as well as the caption. Time 0 marks the

beginning of treatment with the start of the induction phase, which is better displayed with a legend at the bottom marking the three treatment phases of the study. The end of the bars represents the patient ending the trial and ceasing trial-specified therapy. The red dots mark the time of progression of disease.

REVIEWERS' COMMENTS

Reviewer #1 (Remarks to the Author):

I thank the authors for addressing the comments.
I have no further comments.

Reviewer #2 (Remarks to the Author):

The authors have correctly addressed the key points of the critiques and made the corresponding changes in the resubmitted article.

Reviewer #3 (Remarks to the Author):

Thanks for the clarification. The authors addressed my concerns appropriately.